# Natural Radionuclide Levels and Radiological Hazards of Khour Abalea Mineralized Pegmatites, Southeastern Desert, Egypt

Reham M. Abd El Rahman [1], Sherif A. Taalab [2,*], Zainab Z. Al Full [3], Mostafa S. Mohamed [1], M. I. Sayyed [4,5,*], Nouf Almousa [6] and Mohamed Y. Hanfi [1,7,*]

[1] Nuclear Materials Authority, El Maadi, Cairo P.O. Box 530, Egypt; mounir_r79@yahoo.com (R.M.A.E.R.); mostafa24264@hotmail.com (M.S.M.)
[2] Department of Geology, Faculty of Science, Al-Azhar University, Cairo P.O. Box 11884, Egypt
[3] Department of Physics, Taibah University, Almadinah Al-Munawarah 42353, Saudi Arabia; zfull@taibahu.edu.sa
[4] Department of Physics, Faculty of Science, Isra University, Amman 11622, Jordan
[5] Department of Nuclear Medicine Research, Institute for Research and Medical Consultations (IRMC), Imam Abdulrahman Bin Faisal University (IAU), P.O. Box 1982, Dammam 31441, Saudi Arabia
[6] Department of Physics, College of Science, Princess Nourah Bint Abdulrahman University, P.O. Box 84428, Riyadh 11671, Saudi Arabia; nmalmousa@pnu.edu.sa
[7] Institute of Physics and Technology, Ural Federal University, St. Mira, 19, 620002 Yekaterinburg, Russia
* Correspondence: sheriftaalab@azhar.edu.eg (S.A.T.); mabualssayed@ut.edu.sa (M.I.S.); mokhamed.khanfi@urfu.ru (M.Y.H.)

**Abstract:** Arranged from oldest to youngest, the main granitic rock units exposed in Khour Abalea are metagabbros, cataclastic rocks, ophiolitic melange, granitic rocks, pegmatite and lamprophyre dykes. The presence of radioactivity associated with the heavy bearing minerals in construction materials—like granite—increased interest in the extraction process. As it turns out, granitic rocks play an important economic part in the examination of an area's surroundings. The radionuclide content is measured by using an NaI (Tl)-detector. In the mineralized pegmatites, U (326 to 2667 ppm), Th (562 to 4010 ppm), RaeU (495 to 1544 ppm) and K (1.38 to 9.12%) ranged considerably with an average of 1700 ppm, 2881.86 ppm, 1171.82 ppm and 5.04%, respectively. Relationships among radioelements clarify that radioactive mineralization in the studied pegmatites is magmatic and hydrothermal. A positive equilibrium condition confirms uranium addition to the studied rocks. This study determined $^{226}$Ra, $^{232}$Th and $^{40}$K activity concentrations in pegmatites samples and assessed the radiological risks associated with these rocks. The activity concentrations of $^{226}$Ra (13,176 ± 4394 Bq kg$^{-1}$), $^{232}$Th (11,883 ± 5644 Bq kg$^{-1}$) and $^{40}$K (1573 ± 607 Bq kg$^{-1}$) in pegmatites samples (P) are greater than the global average. The high activity of the mineralized pegmatite is mainly attributed to the presence of uranium mineral (autunite), uranophane, kasolite and carnotite, thorium minerals (thorite, thorianite and uranothorite) as well as accessories minerals—such as zircon and monazite. To assess the dangerous effects of pegmatites in the studied area, various radiological hazard factors (external, internal hazard indices, radium equivalent activity and annual effective dose) are estimated. The investigated samples almost surpassed the recommended allowable thresholds for all of the environmental factors.

**Keywords:** pegmatites; activity concentration; radioactive minerals; radiological hazards

## 1. Introduction

Human exposure to ionizing radiation has piqued the public's interest. After all, natural-source radiation accounts for most of the human population's overall radiation exposure [1]. Natural uranium and thorium can be detected in different proportions in all terrestrial materials, depending on the geology of each area [2–4].

The radiological hazard associated with exposure to building materials has received much attention in recent years [5]. Terrestrial radionuclides and their decay products donate to the background radiation value in the environment. The mineral, geochemical and physicochemical factors recreate a role in their environment [5,6].

In addition, the public's radiological influence is a major subject of radioecological research [7,8]. Indeed, many previous investigations have been conducted to report the radiation risk; they found that building materials account for much of the annual dosage of natural radioactivity [7,9,10]. Conducting a radiological effect evaluation for building materials to assess and manage the impact of radioactive chains on the environment is a complicated undertaking. That is because any evaluation must adhere to the requirements of sustainable development. Moreover, the effects of radiation should be evaluated utilizing quantifiable values, which can also be used as input factors when depicting environmental transmission and estimating radiation dose [11,12]. Because people nowadays spend 80% of their time indoors on average, it is critical to determine how much radiation is emitted by the building materials. One can assess the possible radiological threats to people who reside in houses made of these materials by knowing the natural radionuclides concentrations in construction materials [13]. The Khour Abalea area is located in Egypt's Eastern Desert, and it is bounded by the Nugrus thrust fault, a prominent shear zone. The shear zone divides the Ghadir block's ophiolitic and low-grade volcanic complexes to the northeast from high-temperature metamorphic rocks from the Hafafit complex in the southwest [14]. Khour Abalea is considered an unusual case of mineralization that includes economically important rare metals [15]. Inhabited in cataclastic rocks, the pluton core is characterized from the NW orientation by porphyritic biotite granites, distorted biotite granites and two mica granites. In addition, muscovite granites formed the SE half of the pluton [16]. Pegmatites have become more essentially a source of several significant minor and trace elements utilized in modern technology, such as lithium, scandium, cesium, tantalum, niobium, tin and rare earth elements. They also contain quartz, mica, feldspar and a plethora of highly valuable and colourful gem minerals [17,18]. Pegmatites in the Khour Abalea area are categorised into mineralised pegmatites parallel to the foliation (NNW-SSE and dip 10–30 due WSW) and emplaced barren pegmatites (NE-SW and dip nearly vertical) which cut the cataclastic rocks. Except for quartzites, in cataclastic rocks, mineralized pegmatites are intermittent and scattered, ranging in width from 20 cm to 50 cm. They occasionally feature boudins structure, which is found primarily in tectonic zones [19]. Using pegmatite samples collected from Khour Abalea area, this study examined the natural radionuclides distribution and assessed the radiation hazard indices.

## 2. Materials and Methods

### 2.1. Geological Setting

Khour Abalea is located near the lower portions of Wadi Abu Rusheid in Egypt's Eastern Desert. It is about 95 km southwest of Marsa Alam City on the Red Sea shore, passing via Wadi Al Gemal and then Wadi Nugrus (See Figure 1).

An old mineral zone rich in emeralds surrounds this region. It is also accessible by a new asphaltic road that runs between Idfu Marsa Alam Road and Sheikh Shadli further south, between latitudes 24°37′0″ and 24°38′0″ N and longitudes 34°46′0″ and 34°46′40″ E (Figure 2). Khour Abalea's primary rock formations called metagabbros, ophiolitic melange, cataclastic rocks, granitic rocks, lamprophyre dykes and pegmatite are Precambrian rock that can be ordered chronologically from oldest to youngest. Granites of porphyritic biotite are followed by distorted biotite and two mica granites, and the Khour Abalea granitic pluton is extended in NW-SE direction (12 km—length) and photogenic in NE-SW direction (3 km—width) (abundant garnet and kyanite crystals).

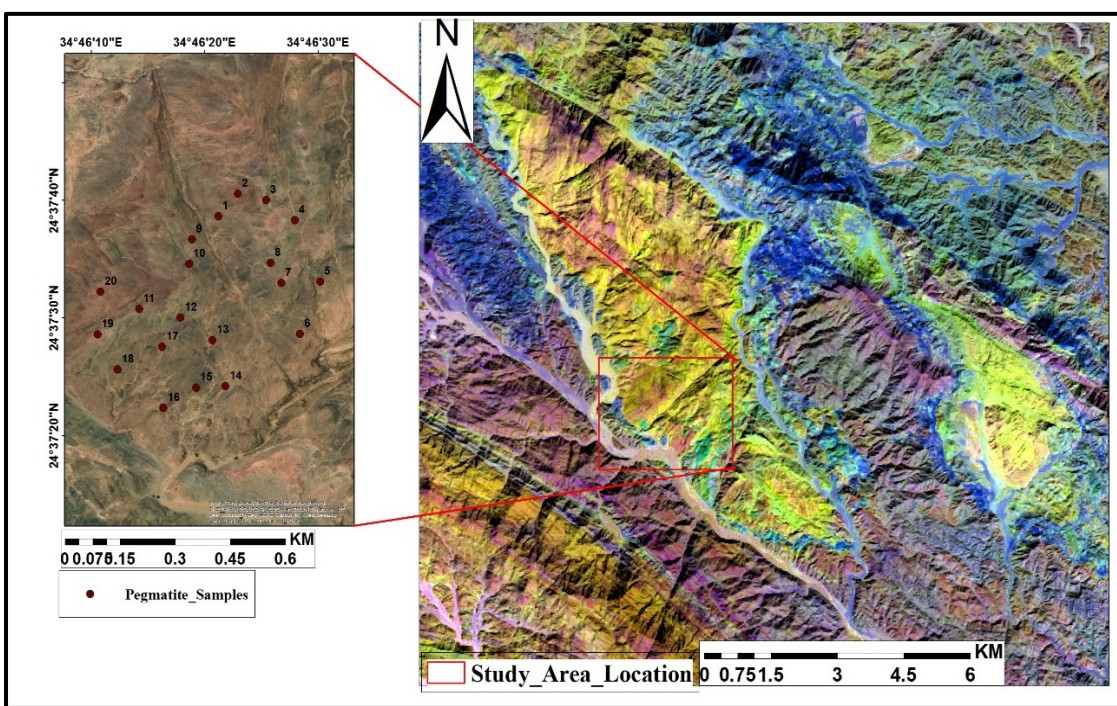

**Figure 1.** Khour Abalea location map and technological samples.

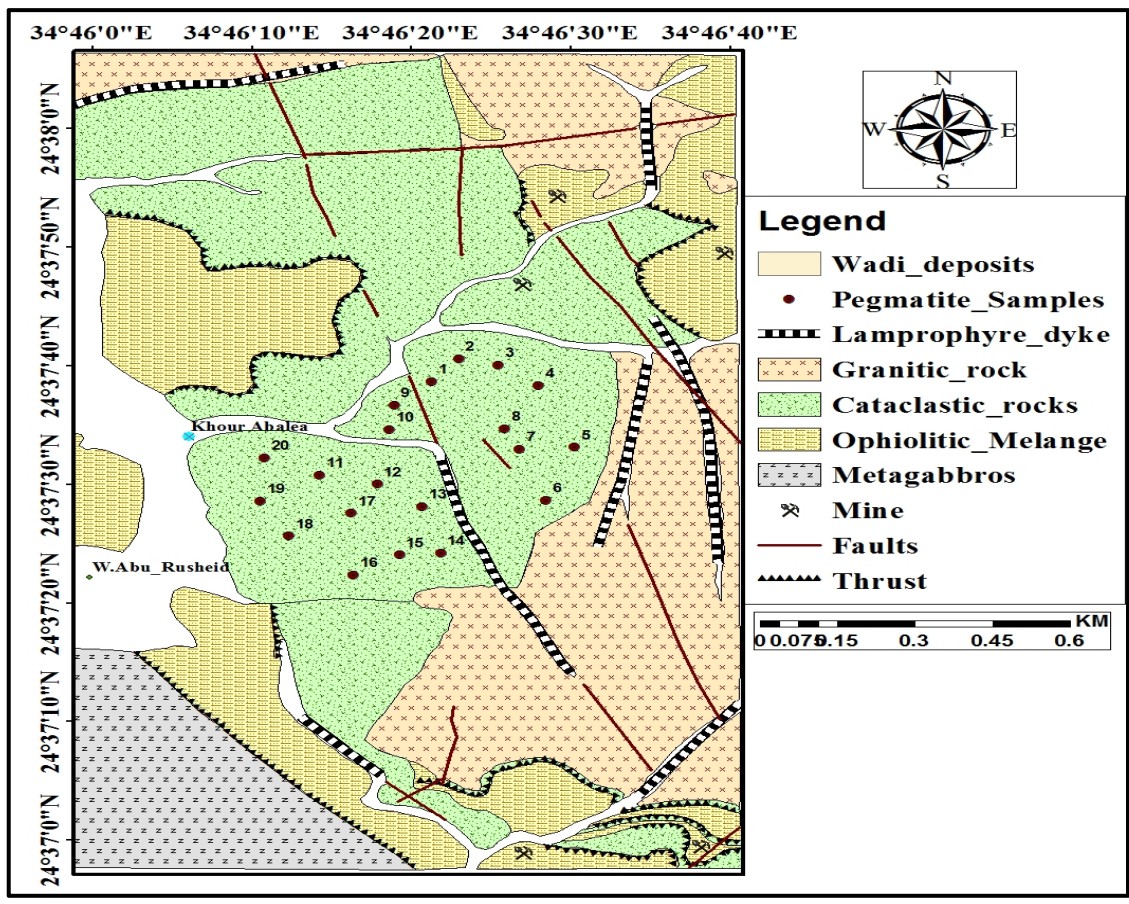

**Figure 2.** A geological map of Khour Abalea area, Southeastern Desert, Egypt [16].

The core of the granitic pluton is made up of cataclastic rocks from the Khour Abalea area. It is enormously sheared, wrapped and foliated (N-S) and is split by two non-parallel shear zones (NNW-SSE and E-W) patterns that span about 2 km$^2$. The shear zones are 2–10 m wide and 400–1500 m long, with a vertical to inclined dip. Shear zones (NNW-SSE) displace E-W shear zones. Discontinuous and brecciated lamprophyre dikes intrude between the two shear zones.

### 2.2. Sampling and Preparation

Pegmatite samples were selected based on the separation process of the mineral portion. The samples were granulated, pulverized, subdivided and segregated to a size fraction of −60 to +120 meshes before being concentrated using a heavy liquids separation procedure with bromoform (specific gravity 2.85 gm/cm$^3$). The magnetite was then separated using a hand magnet. The fractions of the heavy mineral were run through a Frantz Isodynamic Magnetic Separator (Model L-1) with a side tilt of 5° and a forward slope of 20° to segregate the residual magnetite and form multiple magnetic fractions at 0.2, 0.5, 0.7, 1 and 1.5 amperes. Binocular microscopes were used to identify minerals. They were then identified using EDX unit on an environmental scanning electron microscope (ESEM model Philips XL 30).

Twenty pegmatite samples were sent to the lab to be measured for radioactive concentration in order to assess the radiological risks. Each sample weighed about 250 g and was put in clearly labelled polythene bags before being delivered to Cairo's Nuclear and Radiological Regulatory Authority's Radiation Protection Laboratory (Cairo, Egypt) for analysis. The samples were air dried for 72 h at 25 °C at ambient temperature. The dry materials were pulverized, sieved (to a depth of 2000 m) and homogenized extensively. To avoid radon leakage, around $185 \pm 10$ g of the homogenized material was placed carefully into well-labelled plastic containers and properly sealed. The enclosed containers were held for 28 days to accomplish radioactive equilibrium between parents and their nuclide's daughters.

A NaI (Tl) scintillation detector with a crystal that has a square shape with a length of 7.6 cm was applied to estimate the radioelements content (eRa, eTh and K%) in pegmatite samples in ppm. The measurements were identified by the activity concentration in Bq kg$^{-1}$ by applying the conversion factors (11.1 Bq kg$^{-1}$/1ppm, 4.04 Bq kg$^{-1}$/ 1ppm and 313 Bq kg$^{-1}$/% for the aforementioned radioelements respectively) [20]. The detector was set up in a shield in cylindrical lead form with: (diameter = 157 mm), (length = 205 mm) and (thickness = 37 mm) with an attenuation factor of 0.16 for 2.6 MeV $\gamma$-rays, to assure a low background measurement environment. A preamplifier and analyser were parts of the pulse analysis and information analysis equipment, which was connected to a software computer. The $^{226}$Ra, $^{232}$Th and $^{40}$K have gamma energies of 1.764 MeV (I = 15.30%) from $^{214}$Bi, 2.614 MeV (I = 99.754%) from $^{228}$Ac and 1.460 MeV (I = 10.66%) from $^{40}$K [21,22]. Certified reference materials (RGU-1, RGTh-1 and RGK-1) are employed and their densities after becoming pulverized concrete are comparable to those of the construction materials [23]. The vessel was designed such that the radioactivity in samples is dispersed uniformly. The samples are recorded to 2000 s, and the MDAs for $^{226}$Ra, $^{232}$Th and $^{40}$K are 2, 4 and 12 Bq kg$^{-1}$, respectively. Through using the error propagation equation of systematic and random experimental error, the entire uncertainty of radiation data was estimated. The calibration efficiency has systematic inaccuracies of 0.5 to 2%, while the activity values have up to 5% random errors [24].

### 2.3. Radiological Hazards Assessment
2.3.1. Ra$_{eq}$

The Ra$_{eq}$ is a radioactive variable that is widely employed to determine radiation threats to human health. To maintain the public's annual effective dose under 1 mSv, Ra$_{eq}$

values must be less than 370 Bq kg$^{-1}$. The formula below can be used to estimate the Ra$_{eq}$ [25]:

$$Ra_{eq} = A_{(^{226}Ra)} + 1.43\,A_{(^{232}Th)} + 0.077\,A_{(^{40}K)} \tag{1}$$

The value of annual effective dose of 370 Bq kg$^{-1}$ can be achieved when the values of the $^{226}$Ra, $^{232}$Th and $^{40}$K activity concentrations will be 10, 7 and 130 Bq kg$^{-1}$, respectively [26].

### 2.3.2. D$_{air}$ and AED

The absorbed dose rate (D$_{air}$) is a radiometric factor performed to evaluate exposure to the released gamma rays from the radionuclides at distances greater than 1 m from the Earth's surface. To estimate the D$_{air}$ data we used:

$$D_{air} = 0.430\,A_{(^{226}Ra)} + 0.666\,A_{(^{232}Th)} + 0.042\,A_{(^{40}K)} \tag{2}$$

As an outcome, the next Expression (3) was utilised to characterize the accumulated dosages for the public yearly (annual effective dose—AED in several options depending on the exposure is either outdoor or indoor [27,28]).

$$AED_{out}\ (mSv/y) = D_{air} \times 0.2 \times 8760\,h \times 0.7\,Sv/Gy \times 10^{-6}$$

$$AED_{in}\ (mSv/y) = D_{air} \times 0.8 \times 8760\,h \times 0.7\,Sv/Gy \times 10^{-6} \tag{3}$$

### 2.3.3. H$_{ex}$ and H$_{in}$ Indices

To predict the effects of radiation from surface materials on human health, external and internal hazards (H$_{ex}$ and H$_{in}$) are employed. The following Equations (4) and (5) are utilised to compute the hazard index [29,30]:

$$H_{ex} = \frac{A_{(^{226}Ra)}}{370} + \frac{A_{(^{232}Th)}}{259} + \frac{A_{(^{40}K)}}{4810} \leq 1 \tag{4}$$

$$H_{in} = \frac{A_{(^{226}Ra)}}{185} + \frac{A_{(^{232}Th)}}{259} + \frac{A_{(^{40}K)}}{4810} \leq 1 \tag{5}$$

It's not recommended that H$_{ex}$ and H$_{in}$'s principles be ahead of unity [31,32].

### 2.3.4. Annual Gonadal Dose Equivalent (AGDE)

The AGDE has been used to detect D$_{air}$ by general organs, primarily the gonads, on a yearly basis. The AGDE values for $^{226}$Ra, $^{232}$Th, and $^{40}$K are determined based on the gamma radiation emitted [29]:

$$AGDE\ \left(\mu Svy^{-1}\right) = 3.09 A_{(^{226}Ra)} + 4.18 A A_{(^{232}Th)} + 0.314 A_{(^{40}K)} \tag{6}$$

### 2.3.5. Excess Lifetime Cancer (ELCR)

The carcinogenic effects of gamma ray-produced pegmatites can almost certainly be identified if occupants are exposed to them for an extended period of time, both outside and inside. As a result, the ELCR is calculated using the ICRP's cancer risk parameter (RP = 0.05 Sv$^{-1}$) and an outdoor annual effective dose (AEDout) over a lifetime (DL = 70 years) (namely international commission of radiation protection) [33].

$$ELCR = AED \times DL \times RP \tag{7}$$

### 2.3.6. Effective Dose (D$_o$) to Various Body Organs

The following formula is utilized to compute the effective dose rate furnished to a specific organ [34];

$$D_o\ (mSvy^{-1}) = AED \times F \tag{8}$$

where AED is the annual effective dose with two exposure scenarios (outdoor and indoor), and F is the ingestion-to-organ dose conversion factor with values of 0.46, 0.58, 0.62, 0.64, 0.69, 0.82 and 0.68 for the Liver, Ovaries, Kidneys, Lungs, Bone Marrow, Testes and Whole Body, respectively, derived from the ICRP [35].

### 2.4. Multivariate Statistical Analysis (MSA)

These analyses are used to show and emphasize the relationship between radioactive variables, especially the impact of predicted radiological danger parameters on the distribution of radionuclide concentrations in pegmatites. The Varimax normalized technique was used to evaluate the data for PCA. PCA has the advantage of compressing data by reducing the number of dimensions while retaining as much information as feasible. EMB-SPSS Version 21.0 was used to analyse the data statistically.

## 3. Results and Discussion

### 3.1. Mineral Analysis

The high activity of the mineralized pegmatite is at most attributed to the presence of uranium mineral (autunite), thorium minerals and accessories minerals such as monazite, zircon, xenotine, fluorite, samarskite and columbite [36]. In addition to the previously detected minerals, we recorded uranophane [Ca $(UO_2)2(SiO_3OH)_2.5H_2O$], kasolite and carnotite [$K_2$ $(UO_2)$ $(VO_4)_2.3H_2O$].

#### 3.1.1. Uranophane

Uranophane is the most widespread calcium uranium silicate formed primarily in oxidation zones of uranium ore deposits and results from the oxidation-hydration weathering of uraninite in environments rich in $Si^{4+}$ [Ca $(UO_2)_2(SiO_3OH)_2.5H_2O$] [37]. Uranophane crystallizes in the monoclinic system with variable colours such as yellow, lemon yellow, canary yellow, straw yellow and orange–yellow and is transparent to translucent with vitreous to silky lustre. Hardness is 2–3 on the Moho's scale. It is found mostly as fibrous radiating aggregates or felted coatings. Binocular microscope image, SEM analysis and back-scattered electron imaging (BSE) (Figure 3a) indicate that uranophane grains are mainly composed of U (86.80%), Si (3.26%) and Ca (6.53%) and K (2.45%).

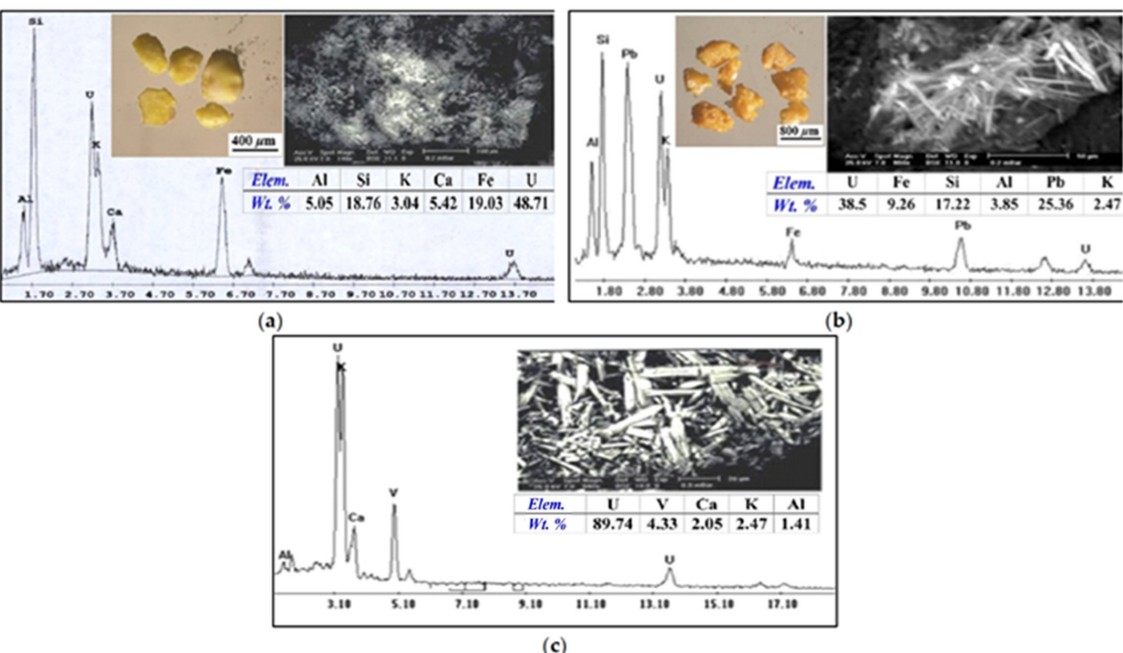

**Figure 3.** (**a**) Binocular microscope images, BSE images and EDX analysis for Uranophane mineral; (**b**) Kasolite mineral; and (**c**) Carnotite mineral.

### 3.1.2. Kasolite

Kasolite is the only identified uranyl silicate containing lead, together with the lead uranyl oxides (Pb $(UO_2)$ $(SiO_4).H_2O$). It is a secondary mineral that forms when meteoric water containing silica reacts with already created secondary uranium minerals. Kasolite is found as collected crystals with a lath-like to needle-like form, with iron oxide staining on occasion. Kasolite is characterized by monoclinic system, bright colours and a resinous or greasy lustre. This mineral is relatively harder when compared to other secondary uranium minerals; the hardness is 4–5 in the Moho's scale. The SEM and BSE (Figure 3b) and the EDX analysis data of kasolite indicate the presence of U (38.5%), Fe (9.26%), Si (17.22%), Al (3.85%), Pb (25.36%) and K (2.47%).

### 3.1.3. Carnotite [$K_2$ $(UO_2)$ $(VO_4)_2.3H_2O$]

Carnotite occurs as rod and needle-like shape crystals (Figure 3c); the EDX analysis data of carnotite indicate the presence of U (89.74%), V (4.33%), Ca (2.05%) and K (2.47%).

### *3.2. Radioelement Distribution*

Radionuclide concentrations of the studied pegmatites are tabulated in Table 1. U, Th, RaeU and K ranged from 326 to 2667 ppm, 562 to 4010 ppm, 495 to 1544 ppm and 1.38 to 9.12% with an average of 1700 ppm, 2881.86 ppm, 1171.82 ppm and 5.04%, respectively. The mean U and Th levels of the examined pegmatites are considerably enriched in both U and Th mean levels 1700.91 ppm and 2881.86 ppm, respectively, as compared to the ranges and averages of Reference [38], (1–6 ppm U and 1–23 ppm Th) [39] and (3 ppm U and 8–17 ppm Th) [40]. eTh/eU ratios in igneous rocks are 3 or 4:1 [41]; however, eTh/eU ratios of the studied mineralized pegmatites have average ratios of 1.7, which is lower than the average of igneous rocks, suggesting uranium enrichment. Some radioelements' inter-relationships could clarify the geochemistry of uranium and thorium. The eTh-eU intercept indicates that the magmatic processes play an essential role in uranium and thorium mineralization; while the eU and eTh relations with eTh/eU show ill-defined trends resulting from the colonization of the pegmatite samples in two groups. The groups include very high and moderate concentrations, which may indicate the presence of more than one pulse of mineralization (Figure 4a–c). A negative correlation of eU and eTh with K may be related to the alteration processes of feldspars pegmatites leading to uranium leaching and thorium adsorption of thorium on the resulting clay minerals leading to the deficiency in U and Th [42,43], (Figure 4d,e). eTh/eU-eTh/K relation manifests in the studied rock samples lying in the moderate sector between fixed and leached zones, indicating partial uranium and thorium mobilization closer to the fixed zone, suggesting they are hazardous if used as construction or decorative materials (Figure 4f).

Authigenic U (Migration Parameter)

The Wignall and Meyers formula (1988) [44] could be used to compute authigenic U enrichment: (Ua) = (U) − (Th)/3.5, where the square brackets signify concentration given in ppm. The estimated uranium values in the mineralized pegmatites are all positive, indicating that the uranium in the investigated granites is authigenic (Table 1).

**Table 1.** Results of radionuclide concentrations of some computed ratios for pegmatite samples.

| S. No. | eU (ppm) | eTh (ppm) | Ra (ppm) | K (%) | eTh/eU | eU/Ra(eU) | eTh/K | eU-eTh/3.5 |
|---|---|---|---|---|---|---|---|---|
| 1 | 2039.00 | 4010.00 | 1491.00 | 4.64 | 1.97 | 1.37 | 864.22 | 893.29 |
| 2 | 2366.00 | 3916.00 | 1509.00 | 5.29 | 1.66 | 1.57 | 740.26 | 1247.14 |
| 3 | 1718.00 | 3796.00 | 1339.00 | 4.55 | 2.21 | 1.28 | 834.29 | 633.43 |
| 4 | 2106.00 | 3746.00 | 1373.00 | 3.91 | 1.78 | 1.53 | 958.06 | 1035.71 |
| 5 | 2244.00 | 3821.00 | 1439.00 | 4.75 | 1.70 | 1.56 | 804.42 | 1152.29 |
| 6 | 1852.00 | 3627.00 | 1340.00 | 7.04 | 1.96 | 1.38 | 515.20 | 815.71 |
| 7 | 1747.00 | 3488.00 | 1259.00 | 5.13 | 2.00 | 1.39 | 679.92 | 750.43 |
| 8 | 1968.00 | 3729.00 | 1356.00 | 1.96 | 1.89 | 1.45 | 1902.55 | 902.57 |
| 9 | 2080.00 | 3576.00 | 1316.00 | 3.80 | 1.72 | 1.58 | 941.05 | 1058.29 |
| 10 | 2180.00 | 3501.00 | 1351.00 | 1.38 | 1.61 | 1.61 | 2536.96 | 1179.71 |
| 11 | 2379.00 | 3870.00 | 1481.00 | 4.22 | 1.63 | 1.61 | 917.06 | 1273.29 |
| 12 | 2265.00 | 3696.00 | 1436.00 | 3.77 | 1.63 | 1.58 | 980.37 | 1209.00 |
| 13 | 2339.00 | 3553.00 | 1393.00 | 5.20 | 1.52 | 1.68 | 683.27 | 1323.86 |
| 14 | 2560.00 | 3751.00 | 1464.00 | 4.18 | 1.47 | 1.75 | 897.37 | 1488.29 |
| 15 | 2667.00 | 3781.00 | 1544.00 | 3.98 | 1.42 | 1.73 | 950.00 | 1586.71 |
| 16 | 426.00 | 629.00 | 512.00 | 7.15 | 1.48 | 0.83 | 87.97 | 246.29 |
| 17 | 419.00 | 579.00 | 594.00 | 8.66 | 1.38 | 0.71 | 66.86 | 253.57 |
| 18 | 404.00 | 628.00 | 541.00 | 5.04 | 1.55 | 0.75 | 124.60 | 224.57 |
| 19 | 342.00 | 570.00 | 508.00 | 6.71 | 1.67 | 0.67 | 84.95 | 179.14 |
| 20 | 326.00 | 562.00 | 495.00 | 9.12 | 1.72 | 0.66 | 61.62 | 165.43 |
| Min | 326.00 | 562.00 | 495.00 | 1.38 | 1.38 | 0.66 | 61.62 | 165.43 |
| Max | 2667.00 | 4010.00 | 1544.00 | 9.12 | 2.21 | 1.75 | 2536.96 | 1586.71 |
| Mean | 1700.91 | 2881.86 | 1171.82 | 5.04 | 1.71 | 1.32 | 828.62 | 880.49 |

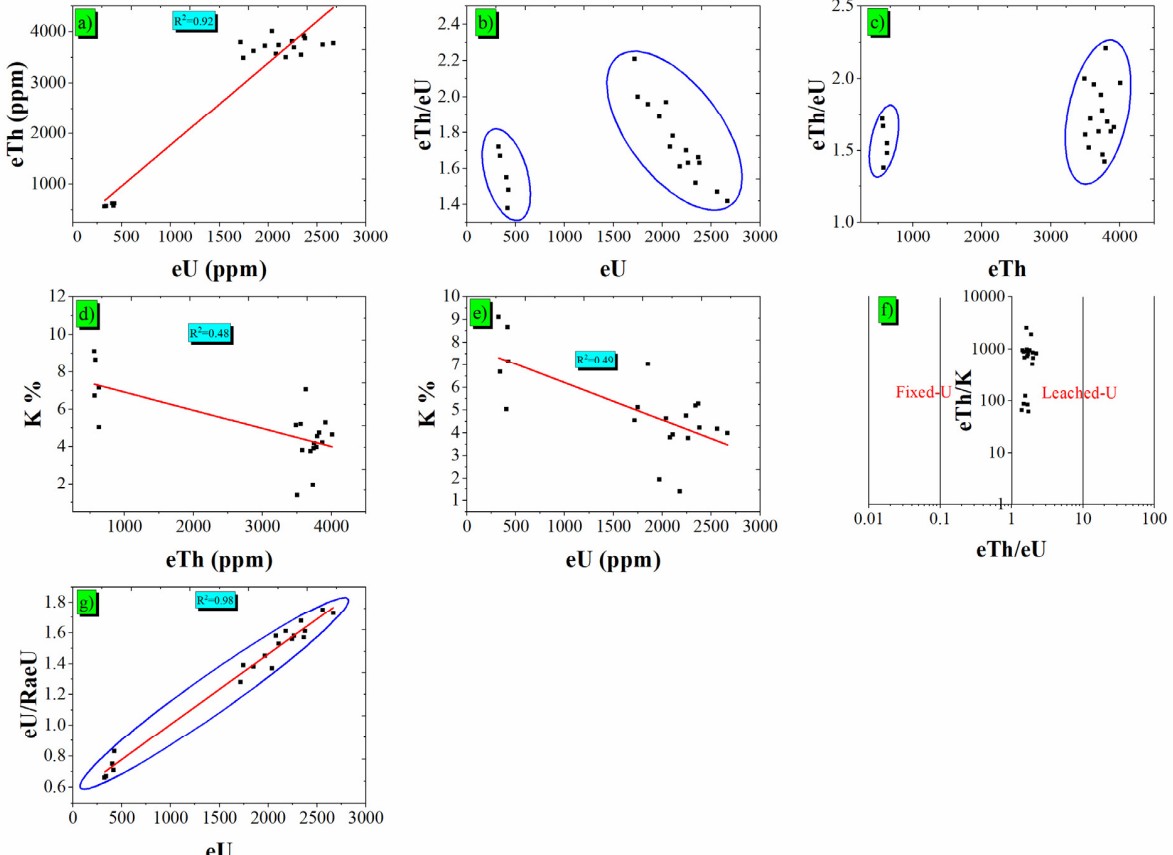

**Figure 4.** Correlations between: (**a**) eTh and eU, (**b**) eTh/eU and eU, (**c**) eTh/eU and eTh, (**d**) K and eTh, (**e**) K and Eu, (**f**) eTh/K and eTh/eU, (**g**) eU/RaeU and eU.

### 3.3. Radioactive Equilibrium

Radioactive equilibrium in the study area can be estimated by the equilibrium factor (P), which was suggested by Reference [45] and applied by [41–43,46]. If the P-factor is ≥1, uranium is being added or removed. The inclusion and removal of uranium are attributed to geological processes such as alteration, which disrupts the equilibrium condition. In addition, groundwater may interact with some uranium deposits, causing uranium to be leached from its initial location and redeposited in new locations. The emission of radon gas, which is one of the uranium products, is another component that affects the equilibrium state. The solubility of radon in water makes it easy to leak through pore spaces, fissures and other fractures. The studied pegmatite samples show disequilibrium conditions where the average of eU/RaeU is 1.32, manifesting uranium addition (Figure 4g and Table 1).

Depending on the radionuclide concentration of the measured samples, radioelements distribution maps were constructed for detecting the sites of anomalous concentrations of U, Th, RaeU ppm and K%, (Figure 5a–d). The distribution maps indicate that the anomalous sites between U, Th and RaeU are matched. The highest values of these elements occur in the west but near the central, northwest and southeast parts of the mapped area, whereas potassium's highest values are located in the east, southeast and southwest.

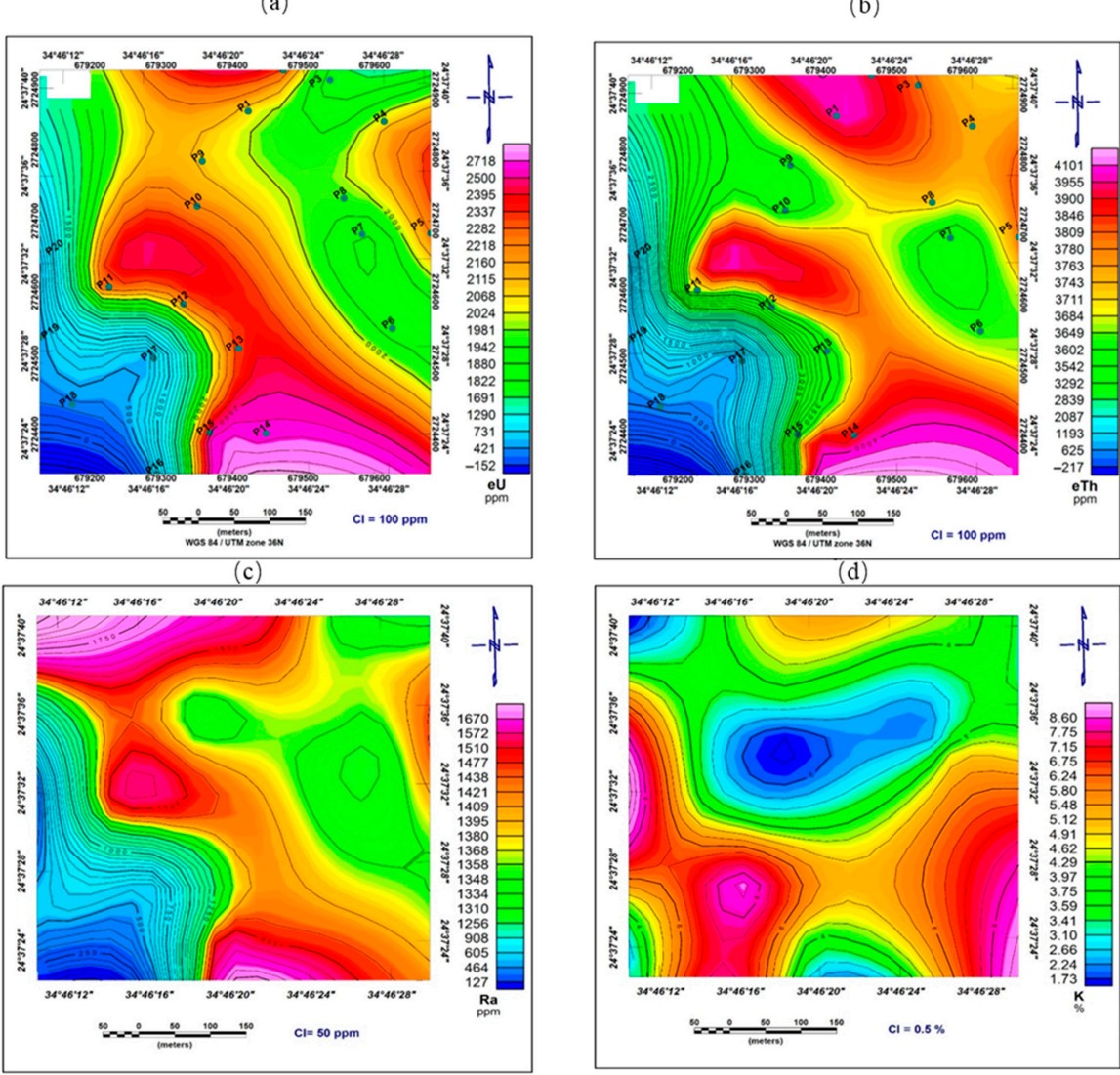

**Figure 5.** Radioelements' distribution maps the accumulation of high concentrations of (**a**) uranium, eU ppm, (**b**) thorium, eTh ppm, (**c**) radium Ra ppm and (**d**) potassium, K%.

### 3.4. $^{226}$Ra, $^{232}$Th and $^{40}$K Activity Concentrations

The measurement data of the $^{226}$Ra, $^{232}$Th and $^{40}$K activity concentrations of 20 collected pegmatites samples are listed in Table 2.

**Table 2.** Activity concentrations of radionuclides in Bq kg$^{-1}$ and the computed environmental parameters in the pegmatite of Khour Abalea area.

| Samples | $^{226}$Ra | $^{232}$Th | $^{40}$K | Ra$_{eq}$ | H$_{in}$ | H$_{ex}$ | D$_{air}$ (nGy/h) | AED$_{out}$ (mSv) | AED$_{in}$ (mSv) | AGDE (mSv) | ELCR $\times 10^{-3}$ |
|---|---|---|---|---|---|---|---|---|---|---|---|
| P1 | 16,550 | 16,200 | 1452 | 39,829 | 152 | 108 | 17,491 | 21.5 | 86 | 119 | 75 |
| P2 | 16,750 | 15,821 | 1656 | 39,501 | 152 | 107 | 17,362 | 21.3 | 85 | 118 | 75 |
| P3 | 14,863 | 15,336 | 1424 | 36,903 | 140 | 100 | 16,188 | 19.9 | 79 | 110 | 69 |
| P4 | 15,240 | 15,134 | 1224 | 36,976 | 141 | 100 | 16,232 | 19.9 | 80 | 111 | 70 |
| P5 | 15,973 | 15,437 | 1487 | 38,162 | 146 | 103 | 16,764 | 20.6 | 82 | 114 | 72 |
| P6 | 14,874 | 14,653 | 2204 | 35,998 | 137 | 97 | 15,813 | 19.4 | 78 | 108 | 68 |
| P7 | 13,975 | 14,092 | 1606 | 34,249 | 130 | 93 | 15,034 | 18.4 | 74 | 103 | 65 |
| P8 | 15,052 | 15,065 | 613 | 36,642 | 140 | 99 | 16,078 | 19.7 | 79 | 110 | 69 |
| P9 | 14,608 | 14,447 | 1189 | 35,358 | 135 | 96 | 15,523 | 19.0 | 76 | 106 | 67 |
| P10 | 14,996 | 14,144 | 432 | 35,255 | 136 | 95 | 15,489 | 19.0 | 76 | 106 | 66 |
| P11 | 16,439 | 15,635 | 1321 | 38,899 | 150 | 105 | 17,092 | 21.0 | 84 | 117 | 73 |
| P12 | 15,940 | 14,932 | 1180 | 37,383 | 144 | 101 | 16,431 | 20.2 | 81 | 112 | 71 |
| P13 | 15,462 | 14,354 | 1628 | 36,114 | 139 | 98 | 15,880 | 19.5 | 78 | 108 | 68 |
| P14 | 16,250 | 15,154 | 1308 | 38,021 | 147 | 103 | 16,714 | 20.5 | 82 | 114 | 72 |
| P15 | 17,138 | 15,275 | 1246 | 39,078 | 152 | 106 | 17,195 | 21.1 | 84 | 117 | 74 |
| P16 | 5683 | 2541 | 2238 | 9489 | 41 | 26 | 4252 | 5.2 | 21 | 29 | 18 |
| P17 | 6593 | 2339 | 2711 | 10,147 | 45 | 27 | 4570 | 5.6 | 22 | 31 | 20 |
| P18 | 6005 | 2537 | 1578 | 9755 | 43 | 26 | 4371 | 5.4 | 21 | 30 | 19 |
| P19 | 5639 | 2303 | 2100 | 9094 | 40 | 25 | 4082 | 5.0 | 20 | 28 | 18 |
| P20 | 5495 | 2270 | 2855 | 8961 | 39 | 24 | 4027 | 4.9 | 20 | 27 | 17 |
| Min. | 5495 | 2270 | 432 | 8961 | 39 | 24 | 4027 | 4.9 | 20 | 27 | 17 |
| Max | 17,138 | 16,200 | 2854.56 | 39,829 | 152 | 108 | 17,491 | 21.5 | 86 | 119 | 75 |
| Average | 13,176 | 11,883 | 1573 | 30,291 | 117 | 82 | 13,330 | 16.3 | 65 | 91 | 57 |
| SD | 4394 | 5644 | 607 | 12,407 | 45 | 34 | 5410 | 6.6 | 27 | 37 | 23 |

Table 3 summarises descriptive statistics of radionuclides activity concentrations in the pegmatites. The radioactive concentrations data, as shown in Table 1, are as follows: The mean $\pm$ SD (Min/Max) activity concentrations of $^{226}$Ra, $^{232}$Th and $^{40}$K are 13176 $\pm$ 4394 (5495/17,138) Bqkg$^{-1}$, 11,883 $\pm$ 5644 (2270/16,200) Bqkg$^{-1}$ and 1573 $\pm$ 607 (432/2855) Bq kg$^{-1}$, respectively. The mean values of Ra-226, Th-232 and K-40 activity concentrations in the pegmatites samples are much greater than the approved worldwide average of 32, 45 and 412 Bqkg$^{-1}$, respectively [4]. Table 3 summarized the descriptive statistical analysis for the acquired activity levels corresponding to $^{226}$Ra, $^{232}$Th and $^{40}$K in the pegmatites samples.

**Table 3.** Statistics of the data corresponding to the activity of radionuclides.

|  | $^{226}$Ra | $^{232}$Th | $^{40}$K |
|---|---|---|---|
| N | 20 | 20 | 20 |
| Mean | 13,176 | 11,883 | 1573 |
| SD | 4394 | 5644 | 607 |
| Min | 5495 | 2270 | 432 |
| Max | 17,138 | 16,200 | 2855 |
| Skewness | −1.14 | −1.22 | 0.44 |
| Kurtosis | −0.57 | −0.52 | 0.42 |
| CV, % | 33 | 47 | 39 |
| GM | 12,208 | 9498 | 1448 |

The frequency of the distribution was determined by analyzing all related radionuclides (histograms are shown in Figure 6a–c). The normal distribution was obtained to [40]K in Figure 6a–c, while [226]Ra and [232]Th distributions were displayed in several multi-modality levels.

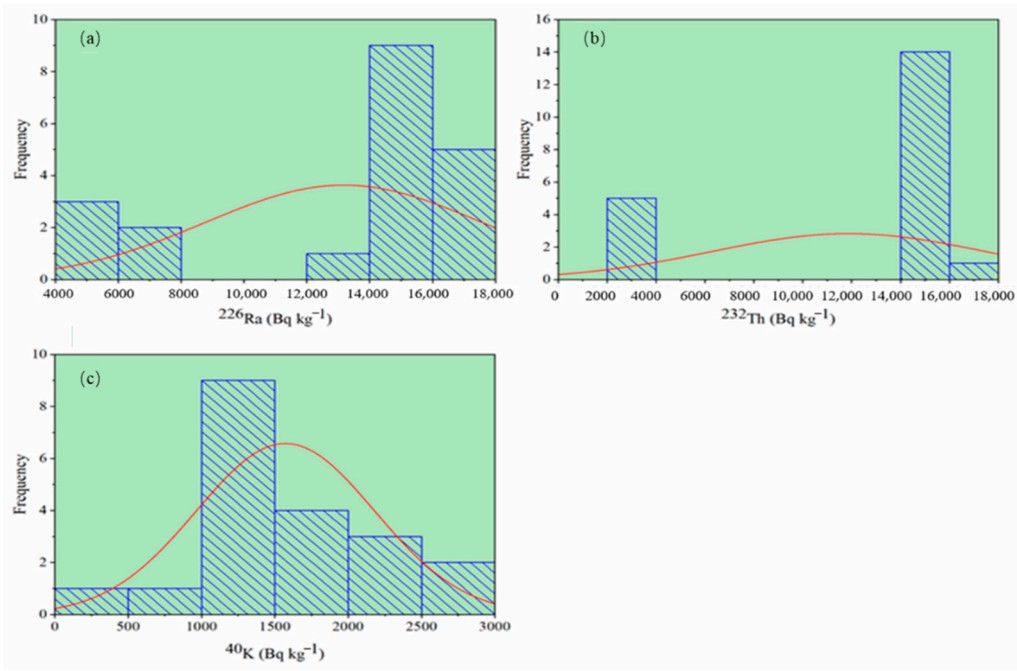

**Figure 6.** The histogram frequency distribution of: (**a**) [226]Ra, (**b**) [232]Th and (**c**) [40]K activity concentrations.

The feature of multimodal for radioelements [226]Ra and [232]Th is owing to the complication of minerals in the pegmatites samples. The confirmation of normal distribution was done by applying the Kolmogorov–Smirnov (KS) test, which illustrates that the distribution of the data was normal (Table 4).

**Table 4.** Results of the Kolmogorov-Smirnov test for normality.

| Radionuclide | KS * | | |
|---|---|---|---|
| | DF | Statistic | *p* |
| [226]Ra | 20 | 0.33 | 0.02 |
| [232]Th | 20 | 0.40 | 0.00 |
| [40]K | 20 | 0.20 | 0.38 |

\* Lilliefors significance correction.

The *p*-value was lower than 5%, implying that the distribution of [226]Ra and [232]Th radionuclides activity in pegmatites is not normal, established on the null hypothesis of KS test. The geological nature of the samples under investigation, which contains uranium-enriched silica and shear zone dykes, is linked to the high content of [238]U [47]. Furthermore, the existence of U and Th in the chemical compositions of heavy minerals like fluorite, zircon, apatite, uraninite and thorianite determined in granitic and basaltic rocks could be responsible for high levels of radioactivity [48,49].

With positive results suggesting the asymmetric distribution, the skewness anticipated the asymmetric distribution, which conformed to the key descriptive statistics of radioelement activity concentrations. Their negative findings allude to the tail of the asymmetric distribution. As a result, negative skewness data for [226]Ra and [232]Th activity concentrations reflect a negative value with the tail of the asymmetric nature, but positive skewness data for [40]K activity concentrations allude a positive asymmetric nature. Second, the kurtosis data show the Preakness of the probability distribution. The radioelements [226]Ra and [232]Th

have negative kurtosis coefficients, which means that their distributions are flat. While the $^{40}$K kurtosis coefficient was positive, indicating that the probability distribution's Preakness was present. Table 3 shows that the standard deviation values for all investigated radionuclides are less than the mean, indicating that the predicted radionuclides in the pegmatites samples have a high level of uniformity. Table 3 shows that the identified radionuclides $^{226}$Ra, $^{232}$Th and $^{40}$K had moderate CV values of 33%, 47% and 39%, respectively. This may be denoted by the presence of introduce radioactive minerals in the pegmatites samples. It was discovered that 75% of the pegmatites samples had higher activity of $^{226}$Ra than the mean value, while 45% of the samples had higher activity of $^{40}$K than the mean value.

The difference among the activity concentrations for the investigated radionuclides with the recent literature is described in Table 5. The present data display much higher than the previous data.

**Table 5.** Comparison of $^{226}$Ra, $^{232}$Th and $^{40}$K activity concentration in the pegmatites samples in Khour Abalea area with numerous studies from the literature from different countries around the world.

| Country | $^{226}$Ra | $^{232}$Th | $^{40}$K | References |
|---------|------------|------------|----------|------------|
| Greek | 74 | 85 | 881 | [24] |
| India | 25.88 | 42.82 | 560.6 | [50] |
| Iran | 77.4 | 44.5 | 1017.2 | [51] |
| Jordan | 41.5 | 58.4 | 897 | [52] |
| Nigeria | 63.29 | 226.6 | 832.5 | [53] |
| Palestine | 71 | 82 | 780 | [54] |
| Spain | 84 | 42 | 1138 | [55] |
| Saudi Arabia | 28.8 | 34.8 | 665.08 | [56] |
| Turkey | 80 | 101 | 974 | [57] |
| Egypt | 137 | 82 | 1082 | [58] |
| Egypt | 13,176 | 11,883 | 1573 | Present study |

### 3.5. Radiological Hazard Impacts

The gamma emitter of gathered pegmatites samples is used to detect the consequences of risk risks using the aforementioned criteria. Table 2 shows the $H_{ex}$ and $H_{in}$ indices, as well as the Raeq, for each pegmatite sample. The $Ra_{eq}$ mean value is 30,291 Bq kg$^{-1}$, which is greater above the specified limit of 370 Bq kg$^{-1}$, and the results in the tested pegmatites samples varied between 8961 and 39,829 Bq kg$^{-1}$.

Raeq's most recent findings show the presence of radium and thorium in pegmatites collections with high activity concentrations. The permitted limit (1) is recognized for all external and internal danger indicators related with the collected pegmatites samples. The $H_{ex}$ values range from 24 to 108, with an average of 82, while the $H_{in}$ data ranges from 39 to 152, with an average of 117. As a result, severe health concerns can be discovered as a result of external gamma radiation exposure [59]. Table 2 shows the associated statistics for the calculated $D_{air}$ data. The maximum $D_{air}$ value (17,491 nGy h$^{-1}$) is significantly greater than the previously reported limit (59 nGy h$^{-1}$) for pegmatites from sample P1. $D_{air}$ data has a statistical range of 4027 to 17,491 nGy h$^{-1}$, with a mean value of 13,330 nGy h$^{-1}$, which is higher than the given limit [4,60]. The AED values for AED$_{out}$ and AED$_{in}$ are enlisted in Table 2.

The AED$_{out}$ data ranged from 4.9 to 21.5 mSv, with a mean of 16.3 mSv, which is greater than the 0.07 mSv reported previously in the literature [4], The AEDin's Min and Max data are 20 and 86 mSv, respectively, with a mean value of 65 mSv, which is over 158 times higher than the UN-SCEAR recommended level of 0.41 mSv. This implies that long-term high-dose exposure can result in cancer, DNA in genes, tissue degradation, or coronary heart disease [61]. The AGDE data range alternated between 27 (P1) and 119 (P20) mSvy$^{-1}$ with the mean value of 91 mSvy$^{-1}$, which much greater than the approved limit of 0.3 mSvy$^{-1}$ (Table 2) [4]. Thus, the pegmatite in the examined area is not suitable for construction

materials or infrastructure use. The Min and Max values of ELCR for the examined pegmatites samples are 0.017 to 0.75, and the mean value is 0.057, which would be 196 times higher than the allowed range (0.00029) [33]. This reveals that long-term exposure to the studied pegmatites can cause cancer in the general populace.

Figure 7 shows the effective dose rate supplied to the various organs, where the $D_O$ calculates how much radiation is accumulated in the different human tissues and organs after a person has been exposed to it for a year. The received dose by the testes organ was observed to be the highest with average values of 13 and 54 $mSvy^{-1}$ for outdoor and indoor, respectively, while the received dose by the liver was the lowest with average values of 8 and 30 $mSvy^{-1}$ for outdoor and indoor, respectively. According to these findings, the projected doses to the various organs investigated are all higher than the acceptable worldwide level dose intake to the body organ of 1.0 mSv yearly. The testes have a higher dose than the other organs, while the liver has lower values of $D_o$, which is justified by the rate of food nutrient absorption [62]. This demonstrates that exposure to gamma rays from the pegmatites at the studied area significantly affects the radiation dosage to these adult organs.

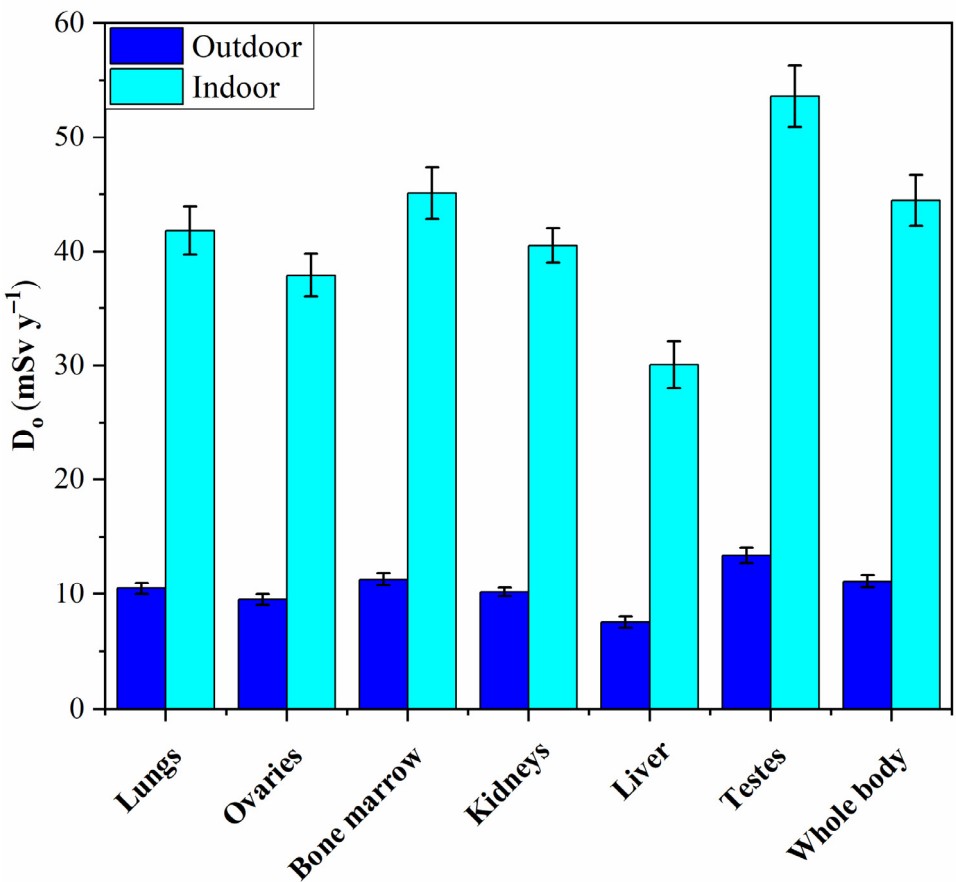

**Figure 7.** The effective dose rate associated with the particular organs from air dose, indoor and outdoor.

### 3.6. Pearson Correlation Matrix (PCM)

The PCM was determined in the present investigation to determine the intensity of the association among the radioactive factors (Table 6). The correlation intensity was divided into four categories as follows: (0.00–0.19), (0.2–0.39), (0.4–0.79) and (0.8–1.00) for weak, moderate, high and extremely strong, respectively [63]. Table 6 shows a high positive ($R^2 = 0.99$) association between $^{226}$Ra and $^{232}$Th. This can be explained according to the existence of two radionuclides in nature simultaneously [64,65]. The moderate

negative correlations are expected between $^{226}$Ra and $^{40}$K ($R^2 = 0.69$) as well as $^{232}$Th and $^{40}$K ($R^2 = 0.70$), demonstrating its origin from several radioactive decay series in nature. Furthermore, the radiological hazard variables have a strong positive and considerable connection with the $^{226}$Ra and $^{232}$Th. This is due to the association of radiological variables with the radionuclides that have been identified in nature as mostly gamma-ray emitting sources.

**Table 6.** The Pearson correlation matrix (PCM) of the radionuclides $^{226}$Ra, $^{232}$Th, $^{40}$K concentrations and the radiological hazard indices.

| | $^{226}$Ra | $^{232}$Th | $^{40}$K | Ra$_{eq}$ | H$_{in}$ | H$_{ex}$ | D$_{air}$ | AED$_{out}$ | AED$_{in}$ | AGDE | ELCR |
|---|---|---|---|---|---|---|---|---|---|---|---|
| $^{226}$Ra | 1 | | | | | | | | | | |
| $^{232}$Th | 0.99 | 1 | | | | | | | | | |
| $^{40}$K | −0.69 | −0.70 | 1 | | | | | | | | |
| Ra$_{eq}$ | 0.99 | 0.99 | −0.69 | 1 | | | | | | | |
| H$_{in}$ | 0.99 | 0.99 | −0.69 | 0.99 | 1 | | | | | | |
| H$_{ex}$ | 0.99 | 0.99 | −0.69 | 0.99 | 0.99 | 1 | | | | | |
| D$_{air}$ | 0.99 | 0.99 | −0.69 | 0.99 | 0.99 | 0.99 | 1 | | | | |
| AED$_{out}$ | 0.99 | 0.99 | −0.69 | 0.99 | 0.99 | 0.99 | 0.99 | 1 | | | |
| AED$_{in}$ | 0.99 | 0.99 | −0.69 | 0.99 | 0.99 | 0.99 | 0.99 | 0.99 | 1 | | |
| ELCR | 0.99 | 0.99 | −0.69 | 0.99 | 0.99 | 0.99 | 0.99 | 0.99 | 0.99 | 1 | |
| AGDE | 0.99 | 0.99 | −0.69 | 0.99 | 0.99 | 0.99 | 0.99 | 0.99 | 0.99 | 0.99 | 1 |

### 3.7. Principal Component Analysis

On the basis of varimax rotations, the PCA proceeded with the matrix correlation between various components. Eigenvectors and eigenvalues describe the major components, and the percent of the variance is explained among the correlation matrix data (Figure 8). The PC1 is differentiated by the strong positive loading of $^{226}$Ra and $^{232}$Th activity concentrations, accounting for 95.42% of the total variance explained. The PC2 was found to explain 4.49% of the overall variance, which is linked to the positive loading of $^{40}$K. As a result, $^{226}$Ra and $^{232}$Th are the major gamma-emitting sources in all pegmatites in the examined area. Furthermore, the mineral analysis verified the existence of radioactive carrying minerals in the tested pegmatites.

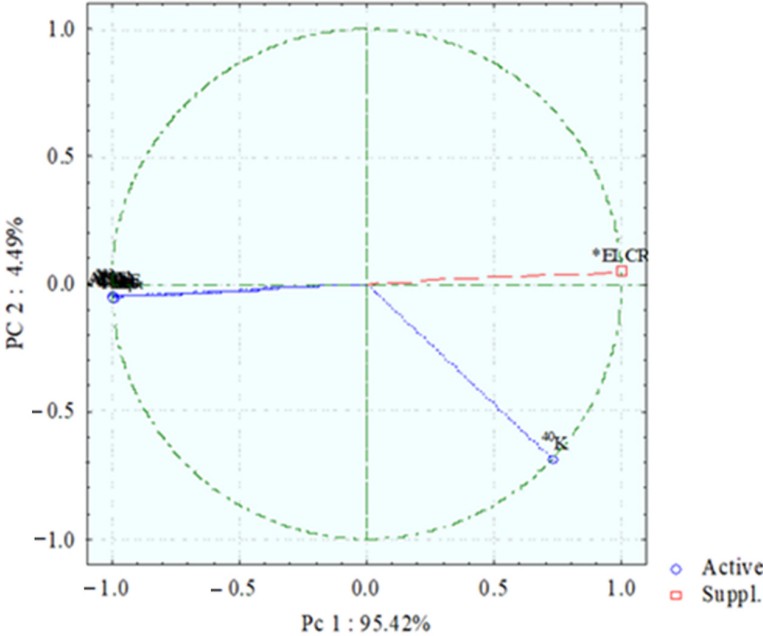

**Figure 8.** Principal component analysis for the radionuclides and radiological factors.

### 3.8. Hierarchical Clustering Analysis

The HCA is used to look at the links among the radiological factors, as shown in Figure 9. In the dendrogram of the examined data, two primary clusters are described. The $^{40}$K is represented by Cluster I, linked to Cluster II, which includes the $^{226}$Ra, $^{232}$Th and the remainder of the radioactive danger factors. This is explained by the radiological hazard characteristics in the pegmatites of the analysed area being connected to the primary gamma radiation sources ($^{226}$Ra and $^{232}$Th). Previous statistical investigations such as Pearson analysis and PCA agree with the HCA.

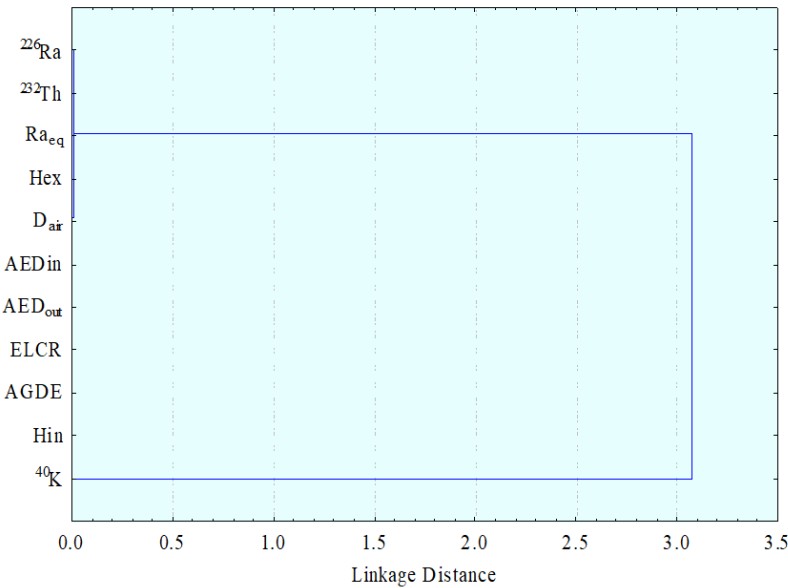

**Figure 9.** Linkage between different statistical radiological hazard indices among the tested samples.

## 4. Conclusions

The radionuclide distribution shows that Abu Rusheid pegmatites are high in uranium high thorium rocks. Relationships among radioelements clarify that the radioactive mineralization in the studied pegmatites is magmatic and hydrothermal. The positive equilibrium condition confirms uranium addition to the studied rocks. Comparisons of the minerals against the other rocks show their enrichment in the radioelements. The radioelements' distribution maps indicate that there is a matching in the anomalous sites between U, Th and RaeU, where the highest values of these elements occur in the west and near to the central, northwest and southeast parts of the mapped area, whereas potassium's highest values are located in the east, southeast and southwest. The obtained data displayed for the $^{226}$Ra, $^{232}$Th and $^{40}$K activity concentrations are 13,176 $\pm$ 4394, 11,883 $\pm$ 5644 and 1573 $\pm$ 607 Bq kg$^{-1}$, respectively, which are higher than the approved worldwide values 33, 45 and 412 Bq kg$^{-1}$. The high activity of the mineralized pegmatite is mainly attributed to uranium mineral (autunite), uranophane, kasolite and carnotite, thorium minerals and accessories minerals as allanite, monazite, zircon, xenotine, columbite, fluorite and samarskite. The radium equivalent content, absorbed dose rate and different radiological hazard variables were estimated. Besides, the values of mean for the annual gonadal dosage equivalent (91 mSvy$^{-1}$) and the raised lifetime cancer risk (ELCR) ($57 \times 10^{-3}$) were estimated. Multivariate statistical strategies are employed and displayed that the $^{226}$Ra and $^{232}$Th principal contributions are predominant in the radioactivity of pegmatites rocks.

**Author Contributions:** Conceptualization, R.M.A.E.R., S.A.T. and M.S.M.; methodology, R.M.A.E.R. and M.S.M.; software, M.Y.H. validation, M.I.S. and Z.Z.A.F.; formal analysis, S.A.T. and M.Y.H.; investigation, Z.Z.A.F., M.S.M. and M.Y.H.; resources, R.M.A.E.R. and S.A.T.; data curation, M.S.M. and M.Y.H.; writing—original draft preparation, R.M.A.E.R., S.A.T., N.A. and M.Y.H.; writing— review and editing, S.A.T. and M.Y.H.; visualization, M.S.M. and Z.Z.A.F.; supervision, M.I.S.; project

administration, R.M.A.E.R. and M.S.M.; funding acquisition, N.A. All authors have read and agreed to the published version of the manuscript.

**Funding:** The authors express their gratitude to Princess Nourah bint Abdulrahman University Researchers Supporting Project number (PNURSP2022R111), Princess Nourah bint Abdulrahman University, Riyadh, Saudi Arabia.

**Institutional Review Board Statement:** Not applicable.

**Informed Consent Statement:** Not applicable.

**Data Availability Statement:** Not applicable.

**Acknowledgments:** The authors express their gratitude to Princess Nourah bint Abdulrahman University Researchers Supporting Project number (PNURSP2022R111), Princess Nourah bint Abdulrahman University, Riyadh, Saudi Arabia.

**Conflicts of Interest:** The authors declare no conflict of interest.

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
