# Peer review of "Natural Radionuclide Levels and Radiological Hazards of Khour Abalea Mineralized Pegmatites, Southeastern Desert, Egypt"

_minerals, doi:10.3390/min12030353_

Round 1
Reviewer 1 Report
Please check the manuscript for all comments.
The presentation of the data is well done.
The English language is good, few mistakes in the last part of the manuscript.

Author Response
Dear Reviewer,
Please find attached the submission of the carefully revised version of the manuscript in Ref., following the minor comments and modification of the Reviewer.
Below is a detailed list of the changes made in response to the Reviewer’s minor comments (in italics), which outlines every change made a point by point. The changes are marked in the manuscript text (yellow highlighted).
Please check the manuscript for all comments.
Response: All corrections are done in the manuscript according to the reviewer’s comments are detected in the attached file.
We thank the Reviewer a lot for the useful and valuable comments that have helped improve the manuscript.
Hoping that all the careful review is sufficient for the direct acceptance of the manuscript, thank you for your time and consideration.
Best wishes,
Mohamed. Y. M. Hanfi
on behalf of all co-authors
Reviewer 2 Report
The manuscript number minerals-1616657 deals with an assessment of natural radionuclide levels and radiological hazards of Khour Abalea mineralized pegmatites, Southeastern Desert, Egypt.
In my opinion the manuscript is quite well written, but the following corrections have to be made before to be considered for publication:
- How are ppm values reported in Tab. 1 obtained? Which instrumentation was employed?
- Improve the Fig. 3 by increasing the characters of the scales.
- Improve the Fig. 5 by inserting the IDs of the investigated sites.
- Check the unity of measurement of ELCR in Eq. (7).
- Line 318: replace Fig. 7 with Fig. 6.
- Line 358: replace Tab. 4 with Tab. 5.
- Line 363: replace Table 1 with Table 2.
- Line 393: insert the reference of the approved value (0.00029).
- Line 414: replace Table 5 with Table 6.
- Improve the Fig. 3 by increasing the characters of the scales.
- Insert a table with the GPS coordinates of the investigated sites.
Author Response
Dear Reviewer,
Please find attached the submission of the carefully revised version of the manuscript in Ref., following the minor comments and modification of the Reviewer.
Below is a detailed list of the changes made in response to the Reviewer’s minor comments (in italics), which outlines every change made a point by point. The changes are marked in the manuscript text (green highlighted).
- How are ppm values reported in Tab. 1 obtained? Which instrumentation was employed?
Response: The samples are measured using the NaI detector in ppm. The following text are added in the manuscript, where it explained the answer of the question " A NaI (Tl) scintillation gamma-ray spectrometer with a crystal size of 76 mm x 76 mm was used to estimate the amounts of radioelements (226Ra, 232Th, and 40K) in pegmatite samples in ppm. The radiological measurements were defined by the activity concentration in Bq kg-1by applying the conversion factors (11.1 Bq kg-1/ 1ppm, 4.04 Bq kg-1/ 1ppm and 313 Bq kg-1/ % for 226Ra, 232Th and 40K, respectively)"
- Improve the Fig. 3 by increasing the characters of the scales.
Response: The figure is improved in the manuscript.
- Improve the Fig. 5 by inserting the IDs of the investigated sites.
Response: The figure is improved in the manuscript and the IDs of the investigated sites are added.
- Check the unity of measurement of ELCR in Eq. (7).
Response: The ELCR has not unit.
- Line 318: replace Fig. 7 with Fig. 6.
Response: Corrected in the manuscript.
- Line 358: replace Tab. 4 with Tab. 5.
Response: Corrected in the manuscript.
- Line 363: replace Table 1 with Table 2.
Response: Corrected in the manuscript.
- Line 393: insert the reference of the approved value (0.00029).
Response: The reference is inserted.
- Line 414: replace Table 5 with Table 6.
Response: Corrected in the manuscript.
- Insert a table with the GPS coordinates of the investigated sites.
Response: The IDs of the investigated sites are added, so the authors believed the required table is not important.
We thank the Reviewer a lot for the useful and valuable comments that have helped improve the manuscript.
Hoping that all the careful review is sufficient for the direct acceptance of the manuscript, thank you for your time and consideration.
Best wishes,
Mohamed. Y. M. Hanfi
on behalf of all co-authors
Round 2
Reviewer 2 Report
I suggest to accept the paper in the present form.